# Grip Force Measurement as a Complement to High-Resolution Ultrasound in the Diagnosis and Follow-Up of A2 and A4 Finger Pulley Injuries

**DOI:** 10.3390/diagnostics10040206

**Published:** 2020-04-08

**Authors:** Xeber Iruretagoiena-Urbieta, Javier De la Fuente-Ortiz de Zarate, Marc Blasi, Felix Obradó-Carriedo, Andoni Ormazabal-Aristegi, Elena Sonsoles Rodríguez-López

**Affiliations:** 1Department of Physical Therapy, Universidad Camilo José Cela, 28692 Madrid, Spain; iruretagoiena.xeber@gmail.com; 2Eskura Osasun Zentroa, 20200 Beasain, Spain; 3Orthopedics Department, Clínica Pakea-Mutualia, 20018 San Sebastián, Spain; jdelafuente@mutualia.eus; 4Plastic Surgery Department, Hospital Universitari Germans Trias i Pujol, 08916 Badalona (Barcelona), Spain; marcblasibrugue@gmail.com; 5Faculty of Health Sciences, Universitat Pompeu Fabra, 08002 Mataró, Spain; fobrado@tecnocampus.cat; 6Ekin Fisioterapia Zentroa, 20550 Arechavaleta, Spain; a.ormazabal@hotmail.com

**Keywords:** annular pulley, ultrasound, strength, diagnosis, tendon–bone distance, rupture, grip, climber

## Abstract

The ability of finger flexors to generate force has been studied in relation to climbing performance. However, not much attention has been paid to the decrease in finger grip force in relation to annular pulley injuries. The purpose of the present study was to determine if an injured annular pulley implies a finger flexor force decrease, as well as its relation to clinical and sonographic changes. We performed an observational study in 39 rock climbers with A2 or A4 pulley injuries to the 3rd or 4th fingers. The variables considered were pain upon palpation, ultrasound tendon–bone distance, and finger grip strength decrease. Three rock climbing grip types were considered: the one finger crimp, open crimp, and close crimp. Injured rock climbers presented a decrease in finger grip strength compared to non-injured controls when performing a one finger crimp (*p* < 0.001). There exists a significant correlation between a tendon–bone distance at the level of the injured pulley and a decreased finger grip strength measured by performing a one finger crimp (*p* = 0.006). A decrease in finger grip strength could be considered in the diagnostic and follow-up process of A2 and A4 pulley injuries to the 3rd and 4th fingers.

## 1. Introduction

Annular finger pulley ruptures, along with finger flexor tendon injuries, are the most frequent injuries experienced by rock climbers, accounting for up to 33% of all injuries in this population [1]. The most frequently injured pulley is the A2, followed by the A4, mainly affecting the 4th finger. Ruptures may be partial or complete and isolated or multiple, the latter being clinically evidenced by the bowstringing of the flexor digitorum superficialis and/or profundus tendons [2,3].

The different types of pulley ruptures may be clinically difficult to differentiate as the clinical presentation is unspecific, unless they present with the bowstringing of the flexor tendons [4]. Symptomatology consists of acute focal pain, which increases upon palpation, and counter-resisted finger flexor maneuvers, occasionally accompanied by inflammatory signs and ecchymosis [5]. Additionally, when pulleys are injured, the biomechanical efficiency of the anatomical structures contributing to finger grip force is decreased [2,6].

The most important factors in the pulley injury mechanism are finger grip force magnitude together with the eccentric load generated against the pulleys as a consequence of the friction between them and the flexor tendons [7]. Therefore, the majority of pulley injuries occur when a drastic increase in finger grip force occurs, such as when a rock climber suddenly loses foot support [3]. The ability to generate finger grip force depends on the type of grip used by the rock climber. The crimp grip is performed with 90–100° proximal interphalangeal flexion and 10–15° distal interphalangeal extension. When the previous grip is performed with a single finger, it is then called a one finger crimp. There also exists other grip types, such as the half crimp and the slope grips, which are much less harmful for finger pulleys compared to the close crimp [8,9,10]. Overall, rock climbers can support 380 newtons (N) per finger. However, a 70 kg rock climber using a one finger crimp may come to support 450 N with a single finger, exceeding the maximum tolerable load, which for an A2 pulley ranges from 380–400 N [1].

In terms of diagnostic imaging, high-resolution ultrasound has been postulated as the gold standard in pulley rupture diagnosis, as it enables the direct visualization of all finger pulleys and the indirect measurement of the consequence of their rupture by means of the flexor tendon–phalangeal bone distance (tendon–bone distance: TBD) [11,12,13,14]. Although controversy still exists [15], the TBD value that has accumulated the most evidence as a threshold to diagnose a complete pulley rupture is 2 mm [3]. 

The main aim of the present study was to determine if an isolated A2 or A4 pulley injury could in turn decrease finger grip strength. Additionally, we sought to determine whether a correlation exists between finger grip strength decrease, ultrasound TBD findings, and clinical symptoms in A2 or A4 pulley injuries, considering different types of classic rock climbing grips. 

## 2. Materials and Methods

We performed a cross-sectional analytical observational study in which we recruited rock climbers with an A2 or A4 pulley injury, who underwent a clinical, a high-resolution ultrasound and a finger grip strength evaluation.

The inclusion criteria were as follows: (i) >15 years old; (ii) >1 year experience in rock climbing; (iii) diagnosed with an A2 or A4 pulley injury of the 3rd or 4th fingers by means of palpation, high-resolution ultrasound, or both; and (iv) a minimum of two weeks and a maximum of six weeks since the time of injury. Those who met at least one of the following were excluded from the study: (i) multiple pulley ruptures A2/A3/A4; (ii) bilateral injuries diagnosed by means of palpation, high-resolution ultrasound or both; or (iii) previous surgical procedures concerning any region from the elbow to the fingers.

Of the 55 rock climbers who met the inclusion criteria, 26 were excluded; 8 had previous surgery of the A2 pulley, 1 underwent surgery due to Dupuytren disease, 1 was diagnosed with a multiple pulley rupture (A2/A3/A4 without bowstringing), and 16 presented with bilateral A2 or A4 pulley injuries of the 3rd or 4th fingers. Finally, 29 rock climbers (22 A2 injuries/7 A4 injuries) entered the study. A control group composed of 10 rock climbers with similar epidemiological characteristics, >1 year of climbing experience, and without any history of injuries to their upper limbs was also included in the study.

The study was conducted in accordance with the ethical standards of the Declaration of Helsinki [16], and the confidentiality of patient data was respected [17]. This study received ethical approval from the Research Ethics Committee of the Universidad Camilo José Cela (Spain, EOPPADME, 22 March 2018). All patients were informed with regards to the aims and procedures of the study and agreed to participate by signing a statement of informed consent. 

The rock climbers were distributed into four groups according to clinical and high-resolution ultrasound findings. Group 1 was composed of 12 rock climbers with pain upon palpation of the injured pulley and a TBD > 2 mm. Group 2 was composed of 10 rock climbers with pain upon palpation of the injured pulley but with a TBD < 2 mm. Group 3 was composed of 7 rock climbers with a TBD > 2 mm but without pain upon palpation of the injured pulley. Group 4 was the control group, composed of 10 rock climbers without pain upon palpation of the injured pulley and a TBD < 2 mm.

All measurements were performed once by seeking consensus between two members of the present research team (XIU and MB). In the event of any discrepancy, a third member (JDF) took further measurements to resolve it. It was not a blinded study as at least one of the researchers had previously performed the clinical examination of the participants. 

### 2.1. Clinical Evaluation 

The most painful point of the injured A2 or A4 was located upon palpation and evaluated symmetrically on the non-injured hand as a control. The result was registered in a binary way; if palpation reproduced pain, the clinical evaluation was considered positive (symptomatic), while if pain was not elicited, it was considered negative (asymptomatic).

### 2.2. High-Resolution Ultrasound Evaluation 

TBD was measured using a Canon Aplio i800 (Canon Medical Systems S.A., Madrid, Spain) equipped with a 24 MHz linear transducer (Figure 1).

Measurements were performed in the longitudinal axis on the level of the injured pulley (A2 or A4, 3rd or 4th fingers) and on the contralateral finger at the same level as a control, in order to calculate the TBD increase percentage. A TBD > 2 mm was considered a positive ultrasound test, while a TBD < 2 mm was considered a negative ultrasound test. To perform such measurements, the finger studied was positioned at a neutral metacarpophalangeal joint position, with 40° of proximal interphalangeal joint flexion and 10° of distal interphalangeal joint flexion. During the evaluation, the ultrasonographer performed a counter-resisted isometric flexion maneuver until maximum TBD was achieved. As it has been proposed in literature, the anatomical landmarks to systematize the measurement point were the middle point of the proximal phalanx for A2 measurements, and the middle point of the middle phalanx for A4 pulley measurements [6].

### 2.3. Finger Grip Strength Evaluation

To measure the finger grip strength, we used the Bindar device, a mechanic and electronic system which consists of a sensor located within the core of a hold, that records the force generated by the rock climber with a precision of 0.1 kg (Figure 2). We quantified, in Newtons, the maximum grip strength at which pain was elicited.

Following the recommendations made by Amca et al. [18], who, through a polynomial regression algorithm, established a relationship between the depth of the hold, the type of grip, and the strength measurement obtained, a 2-cm-deep edge was chosen as the most suitable hold to measure finger grip strength.

The finger grip strength of the affected finger was measured with the modalities of the one finger crimp, open crimp, and close crimp (Figure 2). Next, the same measurements were made on the non-injured contralateral side in order to calculate the strength deficit percentages for each type of grip. The measurement position followed the indications of Balas et al. [19]: sitting down with a 90° upper trunk flexion, feet kept together on a scale (to verify that there was no impulse against the ground at the time of measurement), and a 180° shoulder flexion with full extension of the elbow and wrist.

The rock climbers did not perform any type of exercise 48 h prior to data collection. Before the measurements, the warm up proposed by MacLeod et al. [20] and later by Fryer et al. [21] was performed with a Fingerboard device. It was composed of three exercises: six isometric contractions at 40% maximum strength for 10 s, six contractions at the same intensity for 10 s with a 2 s rest, and two sets of three maximum strength contractions with each hand and a 30 s rest between repetitions. No other type of exercise or stretching was performed prior to the measurements. After a 3-min rest, the rock climbers were asked to perform a maximum strength contraction up to their pain threshold, which was recorded. First, the injured finger was measured using a one finger crimp, followed by measurements of the open crimp and close crimp. Upon each repetition, a maximum force of 5 s was requested [6]. Three attempts were made with each hand for each grip type, and in the case that the highest value was obtained during the third attempt, a fourth repetition was requested. Before each repetition, the importance of stopping the applied force at the first sign of any ache or pain was emphasized to avoid any risk of further pulley injury. In between repetitions using the same grip, a 30 s rest was protocolized. Between changes of grip or side, a 3 min rest interval was assigned. Once the maximum strength up to the pain threshold was recorded for each grip type in the injured hand, the finger grip strength deficit percentage was calculated by performing the same measurements with the contralateral non-injured side and comparing them. Therefore, absolute and relative strength values were not considered, but only finger grip force deficit percentages [19].

### 2.4. Statistical Analysis 

All statistical tests were performed using SPSS 22.0 software (SPSS Science, Chicago, IL, USA). The quantitative outcome measures were assessed through a one-way ANOVA (a Games–Howell test was used for the post hoc analysis). The Levene’s test revealed the lack of homogeneity of variance of the variables, and the Shapiro–Wilk test confirmed their normal distribution. In all tests, the effect sizes were calculated via the partial Eta squared value (η2). All the data are provided as their means, standard deviation, and 95% confidence intervals (CI) when the data are provided as percentages. Finally, Pearson’s correlation was used to analyze the degree of lineal correlation between finger grip strength deficit percentage and TBD. The graphic representation is presented by means of dispersion diagrams. We also determined through a linear regression the relationship between finger grip strength deficit and TBD, as measured by high-resolution ultrasound imaging. All analyses were performed considering a 95% confidence level. The significance was set at *p* < 0.05.

## 3. Results

A total sample of 39 rock climbers, divided into four groups, participated in this study (33 men and six women). Table 1 shows the age and general morphometric parameters of the participants. No differences between groups at baseline in any of the collected variables were found. All the variables followed a normal distribution, except age. Table 2 shows the TBD and strength deficit results.

### 3.1. TBD

When comparing TBD measurements at the level of the A2 or A4 pulleys, a significant group interaction effect was found (F (3,35) = 21.77, *p* < 0.001, η2 = 0.651). Rock climbers diagnosed by ultrasound as having an A2 or A4 pulley injury (TBD > 2 mm) did not show significant differences in TBD (*p* = 0.504) whether they presented with associated pain when palpating (group 1; 2.86 ± 0.58 mm) or not (group 3; 2.51 ± 0.45 mm). Furthermore, no significant differences (*p* = 0.920) were found when comparing rock climbers diagnosed with an A2 or A4 pulley injury only by palpation (group 2; 1.64 ± 0.21 mm) and those without pulley pathology (group 4; 1.71 ± 0.26 mm).

There was no significant group interaction effect (F (3,35) = 2.50, *p* = 0.075, η2 = 0.177) when considering the TBD increase percentage in the multiple comparisons analyzed (*p* > 0.05), except for a significant increase (*p* = 0.007) in Group 1 (pain +, TBD > 2 mm; 201.34 ± 67.5 mm) compared to Group 4 (pain −, TBD < 2 mm; 119.11 ± 11.21 mm).

### 3.2. Grip Strength

A significant group interaction effect was found when comparing amongst groups performing a one finger crimp (F (3,35) = 10.64, *p* < 0.001, η_2_ = 0.477) and, to a lesser extent, when comparing amongst groups performing an open crimp (F (3, 35) = 3.27, *p* = 0.032, η_2_ = 0.219); no significant differences between groups were observed when performing a close crimp (F (3,35) = 1.59, *p* = 0.208, η_2_ = 0.120).

When comparing the grip strength decrease percentage between groups when performing a one finger crimp, significant differences were found between Group 1 (pain +, TBD > 2 mm; 53.99 ± 29.27 N) and Group 3 (pain −, TBD > 2 mm; 16.71 ± 7.17 N; *p* = 0.005) and with Group 4 (pain −, TBD < 2 mm; 8.56 ± 4.74 N, *p* = 0.001). However, no significant differences were found when comparing Group 1 to Group 2 (pain +, TBD < 2 mm; 29.30 ± 21.10 N, *p* = 0.134). No significant differences (*p* = 0.344) were found when comparing Group 2 (pain +, TBD < 2 mm) to 3 (pain −, TBD > 2 mm).

We did not find any significant differences (*p* > 0.005) in the multiple comparisons when performing an open crimp, except for a significant finger flexor force decrease percentage (*p* = 0.049) when comparing Group 1 (pain +, TBD > 2 mm; 21.36 ± 17.52 N) to Group 4 (pain −, TBD < 2 mm; 5.89 ± 4.16 N).

### 3.3. Ultrasound TBD and Grip Strength 

TBD and one finger crimp grip strength deficit percentage are significantly correlated (*p* = 0.006, r = 0.436) (Figure 3). With this type of grip, 19% of the finger grip strength deficit percentage is explained by an altered ultrasound TBD at the level of the injured pulley. TBD does not correlate significantly with finger grip strength deficit percentage either when performing an open crimp or a close crimp.

## 4. Discussion

In the diagnosis of finger pulley injuries, the imaging modality that provides the most information is the high-resolution ultrasound. The following ultrasound signs have been considered for complete pulley ruptures: the hypoechoic swelling of the pulley, a dynamic increase in the flexor tendon–phalanx gap (equivalent to TBD), flexor sheath effusion, and, occasionally, hyperintense injured pulley (color Doppler) [22]. However, to date, the indirect measurement known as TBD has been the most widely considered in scientific literature [3,11,12,13,14,15]. As it is an indirect measure, the specific clinical context is of great value to avoid drawing erroneous diagnostic conclusions. To approach this matter, we here discuss our results in different sub-sections to assess whether the ability to determine finger grip force objectively, the type of grip used when measuring it, the presence or absence of pain, and TBD obtained by ultrasound can together provide relevant information in the diagnosis and/or follow-up of isolated A2 or A4 pulley ruptures.

### 4.1. Grip Strength Measurement 

The results of the present study show relevant differences with respect to a previous study with a similar aim, where the authors did not find a significant association between finger grip strength deficit and finger pulley tears [6]. As we further discuss, methodological differences could account for such differences.

The measurement devices used in the two studies were different. We used a Bindar device, which has been designed specifically in line with the latest indexed literature on rock climber finger grip [23,24], paying special attention to the properties of the hold and to body positioning when performing force measurements [18,19]. The measuring device used by Schöffl et al. [6] is a platform (ErbseÒ, Germany, sampling rate: 1 per millisecond), with which finger grip is not directly measured by means of a sensor inside the hold, as in the Bindar, but it is measured indirectly through the change in the body mass measured on a platform when performing a suspension grabbing on a hold.

Positioning during finger grip strength measurement might be another reason. We chose the sitting position described by Balas et al. [19], in which a concentric-isometric force is exerted during the measurement, measuring only finger grip strength. On the contrary, Schöffl et al. [6] used a hanging with the body in the upright position, which most resembles a real rock climbing situation based on an isometric-eccentric force, but could not be as specific as it does not only measure finger grip strength.

As for the formulas used to obtain the finger grip strength deficit value, in the current study, the absolute strength value was not divided by the body mass to obtain the relative strength, as this penalizes heavier rock climbers [25]. Furthermore, body mass index was not taken into consideration, as in a recent retrospective study it was demonstrated that it is not associated with injuries having occurred within the previous six months as the body can adapt to the load despite the high body mass index [26]. On the other hand, in the formulas used by Schöffl et al. [6], the dominant hand was considered, which we did not. In further studies, both should be considered when choosing the formulas to measure finger grip strength, avoiding relative force measurements and considering whether the hand measured is the dominant one.

The grip types used during force measurement also differed across both studies. Schöffl et al. [6] studied the crimp and slope grips, whereas, in the present study, we did not consider the slope grip, as pulley ruptures are associated with crimp grips [27].

### 4.2. Grip Strength Deficit

We found a significant moderate correlation between TBD at the level of the injured pulley (A2 or A4) and finger grip strength when performing a one finger crimp. This means that a climber with a ruptured A2 or A4 pulley who undergoes an ultrasound diagnosis with a TBD > 2 mm will have a loss in one finger crimp grip strength as well. However, this correlation was not significant when performing an open crimp, nor when performing a close crimp. This could be due to the fact that the ability to generate an effective force is compensated for by the non-injured fingers in multiple finger grips, which cannot occur in single finger grips. All in all, this finding reinforces theories that attribute an initial strength deficit to biomechanical alterations of the finger pulley system. It has been theorized that when a pulley ruptures, the flexor tendons separate from the phalanges, reducing the proximal–distal attachment distance of the flexor digitorum superficialis and/or profundus muscles. As the tendons maintain their length, the total muscle bellies are shortened, in turn decreasing their ability to shorten during muscle contraction and thus their ability to generate force [6]. There exists another theory, also in line with our findings and with a similar approach to that previously mentioned, which is based on a biomechanical formula that considers the amount of finger flexion to be proportional to the longitudinal tendon excursion and the length of the moment arm (degree of finger flexion = tendon excursion/moment arm) [2].

A controversy exists in the literature regarding the TBD threshold to diagnose an A2 or A4 pulley tear, with the most widely accepted measurement at the moment being 2 mm [3]. Taking this value as a reference, our study quantified that this corresponds to a finger grip strength deficit percentage equivalent to 41.4%, taking the contralateral limb as a control in force measurement when performing a one finger crimp. However, this must be interpreted cautiously as the possibility of physiological differences in strength between the upper extremities, for instance, depending on hand dominance, could influence its interpretation in terms of being a physiological versus pathological difference, or partly explained by both [28,29].

### 4.3. Ultrasound TBD, Grip Strength Deficit, and Clinical Signs

When integrating the three diagnostic variables considered in this study, we found three patterns with possible clinical significance. The group of rock climbers with the greatest loss of strength had a TBD > 2 mm on high-resolution ultrasound and presented with pain upon clinical examination (Group 1). The second group with the greatest deficit in strength did not have a TBD > 2 mm on high-resolution ultrasound but did present pain upon clinical examination (Group 2). No significant differences in finger grip strength deficit percentage were found between these two groups when measured performing a one finger crimp. It could be that the rock climbers in Group 2 did not have complete pulley tears but minor or partial tears, preserving part of the pulley integrity and therefore not showing a significant increase in TBD as to be considered pathological. In these patients, it would not be possible to explain the decrease in finger grip strength by the biomechanical theory mentioned previously. However, it would be in line with the fact that pain itself can inhibit muscle contraction via central nervous system control mechanisms [30].

There exists a third group of rock climbers who presented a TBD > 2 mm on high-resolution ultrasound but no pain upon palpation of the affected pulley (Group 3). Once again, several reasons could explain this incongruence between the ultrasound pathological findings and no pain upon clinical examination. It could be attributed to the fact that the clinical symptoms were of shorter duration and intensity. It could also be possible that the ultrasound finding was due to a longstanding pulley rupture or response to submitting the pulley to a repeated stress, in the form of non-pathological pulley slacking adaptation. The latter would also explain why the finger flexor force deficit was lower in this third group when compared to the two previously mentioned (Group 1 and 2) [31].

All in all, the present study suggests that, in most A2 or A4 isolated pulley injuries, there exists an accompanying, to a greater or lesser degree, loss of finger grip strength without a mandatory ultrasound–clinical correlation. Moreover, we have put forward that the presence of pain upon palpation per se does not imply an increase in TBD > 2 mm. This makes it possible to ensure that, although the sum of a pathological TBD on high-resolution ultrasound together with painful symptomatology presents with the greatest values of finger grip strength deficit percentage, there are two other patterns with a probable pathological and/or adaptive background that must be elucidated in future studies to better contextualize TBD measurements and allow better differential diagnosis.

Thus, in A2 or A4 complete pulley ruptures, one finger crimp grip strength measurement is a useful evaluation tool to obtain a more complete diagnosis, along with other well-known diagnosis methods, such as clinical outcome (mechanism of injury, pain perception, range of movement of the finger by Buck-Gramcko score [32], motion pattern, etc.) and ultrasound scanning. Moreover, grip strength measurement is relevant during the patient follow-up, as improvement in strength could be a useful indicator in patient progress [33].

The main limitations of this study were the low number of subjects included, lack of information about the climbing ability of the subjects, not differentiating subjects by gender, not differentiating A2 from A4 pulleys, and not considering the dominant hand in force measurements. Moreover, we only took into account TBD as a sonographic outcome, further studies should include other ultrasound signs to correlate them to TBD in each of the clinical or subclinical patterns found in the present study.

## 5. Conclusions

A2 or A4 pulley injuries generate a significant finger grip strength deficit when performing a one finger crimp. A finger grip strength deficit percentage >41.4% when performing a one finger crimp, as measured by a Bindar, is equivalent to a high-resolution ultrasound TBD > 2 mm. Finger grip strength measurement allows us to complement the diagnosis and follow-up of patients with A2 or A4 pulley injuries. This study suggests that the diagnosis of finger pulley injuries is more complete if clinical signs, ultrasound findings, and finger grip strength are considered together. Data are available upon reasonable request.

## Figures and Tables

**Figure 1 diagnostics-10-00206-f001:**
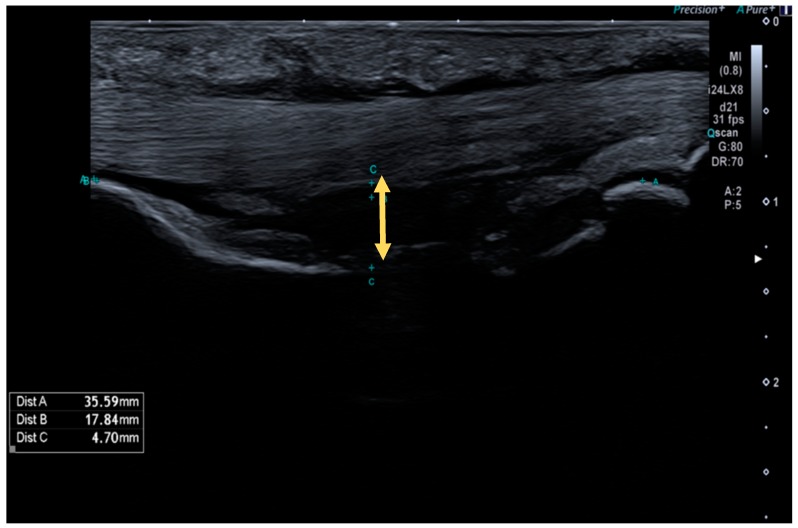
High-resolution ultrasound image of a complete A2 pulley rupture. The arrow indicates the tendon–bone distance (TBD) in the midpoint of the proximal phalange.

**Figure 2 diagnostics-10-00206-f002:**
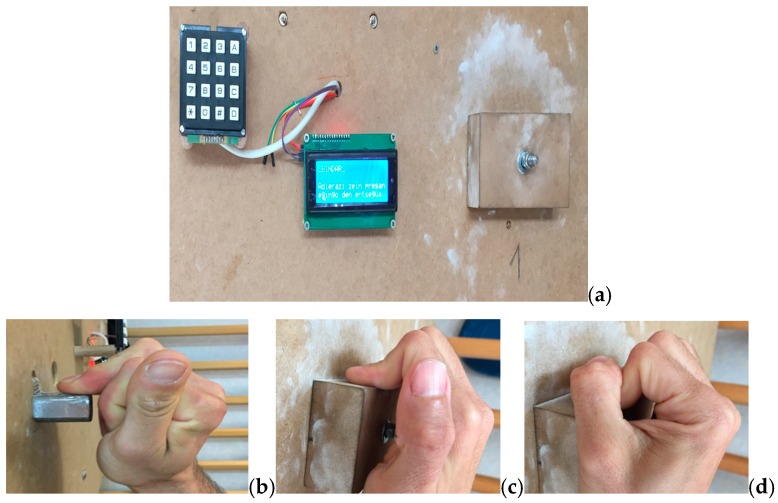
Bindar and grip types. (**a**) Bindar device. (**b**) One finger crimp. (**c**) Open crimp. (**d**) Close crimp.

**Figure 3 diagnostics-10-00206-f003:**
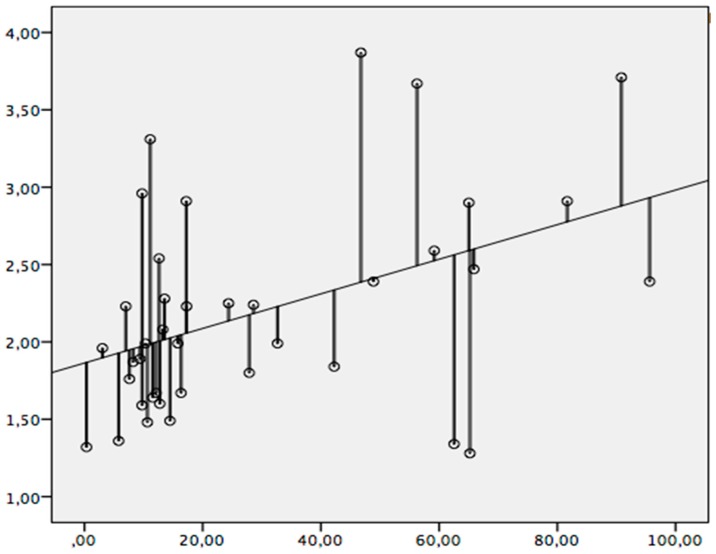
Significant correlation between TBD and finger grip strength deficit percentage when performing a one finger crimp. X axis: the percentage of force deficit of the injured finger measured with a Bindar in Newtons. Y axis: the TBD of the injured finger measured in the longitudinal axis in millimeters.

**Table 1 diagnostics-10-00206-t001:** Baseline epidemiological and general morphometric parameters.

Characteristics	Group	*n*	Mean	SD—(95%CI)
Age (years)	Total sample	39	33.46	8.94 (30.56–36.36)
	1 (pain +, TBD +)	12	35.75	9.12 (29.95–41.55)
	2 (pain +, TBD −)	10	34.20	8.95 (27.79–40.61)
	3 (pain −, TBD +)	7	31.29	9.06 (22.90–39.67)
	4 (pain −, TBD −)	10	31.50	9.22 (24.90–38.10)
Weight (kg)	Total sample	39	68.15	10.46 (64.76–71.55)
	1 (pain +, TBD +)	12	68.75	10.05 (61.70–73.70)
	2 (pain +, TBD −)	10	67.70	8.39 (61.70–73.70)
	3 (pain −, TBD +)	7	75.71	10.24 (66.24–85.19)
	4 (pain −, TBD −)	10	62.60	10.89 (54.81–70.39)
Height (cm)	Total sample	39	174.62	7.09 (172.32–176.92)
	1 (pain +, TBD +)	12	175.33	6.58 (171.15–179.52)
	2 (pain +, TBD −)	10	173.80	6.54 (169.12–178.48)
	3 (pain −, TBD +)	7	177.14	4.22 (173.24–181.05)
	4 (pain −, TBD −)	10	172.80	9.73 (165.83–179.77)

* SD, standard deviation; CI, confidence interval; +, positive test; −, negative test.

**Table 2 diagnostics-10-00206-t002:** TBD and finger grip strength deficit percentages.

Variable	Group	*n*	Mean	SD—(95%CI)	*p*-Value *
**Ultrasound results**					
TBD A2/A4 (mm)	1 (pain +, TBD +)	12	2.86	0.58 (2.48–3.23)	<0.001
	2 (pain +, TBD −)	10	1.64	0.21 (1.48–1.79)	
	3 (pain −, TBD +)	7	2.51	0.45 (2.09–2.93)	
	4 (pain −, TBD −)	10	1.71	0.26 (1.51–1.90)	
TBD increase % in comparison to the contralateral non-injured side	1 (pain +, TBD +)	12	201.34	67.50 (158.45–244.23)	*p* = 0.075
	2 (pain +, TBD −)	10	160.44	110.88 (81.11–239.76)	
	3 (pain −, TBD +)	7	155.66	44.50 (114.50–196.81)	
	4 (pain −, TBD −)	10	119.11	11.21 (111.09–127.13)	
**Force deficit percentage results for each of the three grip types**					
Finger grip strength deficit % using a one finger crimp	1 (pain +, TBD +)	12	53.99	29.27 (35.39–72.58)	<0.001
	2 (pain +, TBD −)	10	29.30	21.10 (14.21–44.40)	
	3 (pain −, TBD +)	7	16.71	7.17 (10.08–23.35)	
	4 (pain −, TBD −)	10	8.56	4.74 (5.17–11.95)	
Finger grip strength deficit % using a close crimp	1 (pain +, TBD +)	12	19.95	20.40 (6.99–32.91)	*p* = 0.208
	2 (pain +, TBD −)	10	9.33	21.37 (−5.95–24.62)	
	3 (pain −, TBD +)	7	4.37	3.83 (0.82–7.92)	
	4 (pain −, TBD −)	10	10.49	6.85 (5.58–15.39)	
Finger grip strength deficit % using an open crimp	1 (pain +, TBD +)	12	21.36	17.52 (10.23–32.50)	*p* = 0.032
	2 (pain +, TBD −)	10	10.04	14.34 (−0.20–20.30)	
	3 (pain −, TBD +)	7	7.90	6.45 (1.93–13.86)	
	4 (pain −, TBD −)	10	5.89	4.16 (2.91–8.88)	

* one way ANOVA. SD, standard deviation; CI, confidence interval; +, positive test; −, negative test.

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
