# Peer review of "Grip Force Measurement as a Complement to High-Resolution Ultrasound in the Diagnosis and Follow-Up of A2 and A4 Finger Pulley Injuries"

_diagnostics, 2020, doi:10.3390/diagnostics10040206_

Round 1
Reviewer 1 Report
Thank you for a very well written paper!
In this observational study of 29 rock climbers, the authors seek to determine if an isolated A2 or A4 pulley injury cour decrease finger grip strength, and correlate to ultrasound TBD findings.
The authors included adults (>15 years) with >1 yr experience in rock climbing and injury to A2 or A4 pulley of 3rd or 4th finger within the last 2 to 6 weeks. Exclusion criteria were multiple or bilateral ruptures and previous surgery. Of 55 screened, 29 were eligible. These were again distributed in four smaller groups according to Tendon-Bone-Distance (TBD) > or < 2mm and pain status yes/no.
Finger grip strength and ultrasound examinations are nicely described alongside with images. No reliability exercise was performed or referred to.
The main finding is that TBD and finger grip strength deficit percentages are significantly correlated when measuring strength deficit according to a one-finger crimp, but not open crimp or close crimp. Furthermore, there seems to be a loss of finger grip strength without a mandatory ultrasound-clinical correlation. This shows that finger grip strength complements the diagnosis and follow-up of patients with an A2 or A4 pulley injury.
The authors are finally open about the limitations of the study, which should not be a hinder for publication.
Major comment: I am not sure I would write in the final line of the study that the diagnosis is "more reliable". This study does not test whether the diagnosis of pulley injuries is done more correctly with the addition of finger strength. Certainly, it gives a more complete assessment, as also indicated in the conclusion. Suggestion: switch from "more reliable" to "more complete".
Very minor:
- Introduction: rewrite "The different types of pulley ruptures may be clinically difficult to differentiate as the clinical presentation is unspecific, unless they present with bowstringing of the flexor tendons4.
- page 2 third last paragraph "had had"
- last paragraph method: "was took"?
Author Response
Response to Reviewer 1 Comments
April 5, 2020
Dear Reviewer,
Please find uploaded a new version of our manuscript “Grip Force Measurement as a Complement to High-Resolution Ultrasound in the Diagnosis and Follow-Up of A2 and A4 Finger Pulley Injuries”, which has been revised according to the comments made. Overall, we found these comments useful and feel they have improved the description of our work. All the changes made have been indicated using the "Track Changes" function. Below we provide replies to each of the points raised.
Thank you for the time devoted to the correction of our study and for all your suggested comments. Thanks to them, we have been able to improve our work.
Yours sincerely,
Comments and Suggestions for Authors
Thank you for a very well written paper!
In this observational study of 29 rock climbers, the authors seek to determine if an isolated A2 or A4 pulley injury cour decrease finger grip strength, and correlate to ultrasound TBD findings.
The authors included adults (>15 years) with >1 yr experience in rock climbing and injury to A2 or A4 pulley of 3rd or 4th finger within the last 2 to 6 weeks. Exclusion criteria were multiple or bilateral ruptures and previous surgery. Of 55 screened, 29 were eligible. These were again distributed in four smaller groups according to Tendon-Bone-Distance (TBD) > or < 2mm and pain status yes/no.
Finger grip strength and ultrasound examinations are nicely described alongside with images. No reliability exercise was performed or referred to.
The main finding is that TBD and finger grip strength deficit percentages are significantly correlated when measuring strength deficit according to a one-finger crimp, but not open crimp or close crimp. Furthermore, there seems to be a loss of finger grip strength without a mandatory ultrasound-clinical correlation. This shows that finger grip strength complements the diagnosis and follow-up of patients with an A2 or A4 pulley injury.
The authors are finally open about the limitations of the study, which should not be a hinder for publication.
Response: thank you very much for your revision as it is very accurate and very well summarized. You reflected our aims, methods, findings and limitations exactly as they are.
Major comment: I am not sure I would write in the final line of the study that the diagnosis is "more reliable". This study does not test whether the diagnosis of pulley injuries is done more correctly with the addition of finger strength. Certainly, it gives a more complete assessment, as also indicated in the conclusion. Suggestion: switch from "more reliable" to "more complete".
Response: about your major comment we definitely agree with you and have made the change suggested. We are aware that this study puts forward that taking into account finger strength gives additional information to the clinician, however we have not quantified the realiability (intra/inter) or specific weight it should have in the diagnostic work-up.
Very minor:
- Introduction: rewrite "The different types of pulley ruptures may be clinically difficult to differentiate as the clinical presentation is unspecific, unless they present with bowstringing of the flexor tendons4.
Response: change made.
- page 2 third last paragraph "had had"
Response: change made.
- last paragraph method: "was took"?
Response: change made.
Reviewer 2 Report
Summary of study finding
The authors investigated the relation between ultrasound findings of A2 or A4 injury to the 3rd and 4th finger (assessment of bowstring of FDS, FDP) and 3 types finger grip strengths in 39 rock climbers and found the positive correlation between the severity of bowstring of FDS, FDP and the degree of one finger crime strength deficit in 39 rock climbers. I think that the result of this study was adequate, however, I think the authors need some work and revise the manuscript.
Major comments
Title
Accurately reflects content of the manuscript.
Abstract
It can be understood without the content of the manuscript.
Introduction
Adequate and relevant.
Methods
- US assessment
Did the member who performed US assessment know the clinical symptom or under the blinded condition to clinical symptom?
Did the author assess the intratester reliability and intertester reliability for ultrasonographic examination of TBD?
It seems to be useful to assess the inflammation of the soft tissue by Doppler ultrasound.
Why did the authors evaluate the inflammation of finger by Doppler ultrasound in this study?
Results
Did the author assess the BMI among four group?
Discussion
This study demonstrated the usefulness of indirect evaluation of A2 or A4 injury (to assess the tendon-bone distance). I think that it seems to be possible to detect A2 or A4 rupture (anatomical rupture) directly with ultrasound. Are there any references related with direct evaluation of A2 or A4 rupture by ultrasonography?
I think the authors need some work on the discussion. If the authors compare other studies to those that are only directly comparable to their own, I believe this study will have a much more powerful and clear paper.
What is the clinical relevances from the result derived from this study for the treatment of A2 or A4 rupture ?
Author Response
Response to Reviewer 2 Comments
April 5, 2020
Dear Reviewer,
Please find uploaded a new version of our manuscript “Grip Force Measurement as a Complement to High-Resolution Ultrasound in the Diagnosis and Follow-Up of A2 and A4 Finger Pulley Injuries”, which has been revised according to the comments made. Overall, we found these comments useful and feel they have improved the description of our work. All the changes made have been indicated using the "Track Changes" function. Below we provide replies to each of the points raised.
Thank you for the time devoted to the correction of our study and for all your suggested comments. Thanks to them, we have been able to improve our work.
Yours sincerely,
MAJOR COMMENTS
Title
Accurately reflects content of the manuscript.
Response: thank you very much.
Abstract
It can be understood without the content of the manuscript.
Response: ok.
Introduction
Adequate and relevant.
Response: thank you.
Methods
- US assessment
Point 1. Did the member who performed US assessment know the clinical symptom or under the blinded condition to clinical symptom?
Response: we have re-written the sixth paragraph of methods to clarify (line 113):
“All measurements were performed once by seeking consensus between two members of the present research team (SIU and MB). In the event of any discrepancy, a third member’s opinion (JDF) was also considered. It was not a truly blinded study as at least one of the researchers had previously performed the clinical examination of the participants.”
Point 2. Did the author assess the intratester reliability and intertester reliability for ultrasonographic examination of TBD?
Response: No. All measurements were performed by one of the authors, however, taking into consideration the opinion of at least one or two (if discrepancy existed with the first). This is why intra o intertester reliability could not be quantified.
Point 3. It seems to be useful to assess the inflammation of the soft tissue by Doppler ultrasound. Why did the authors evaluate the inflammation of finger by Doppler ultrasound in this study?
Response: We did not evaluate inflammation of the soft tissue by Doppler ultrasound. We did not consider it because after going through existing literature about A2 and A4 pulleys diagnosis, it did not appear as a usual outcome measure in pulley rupture studies. However, as some authors have published, the degree of vascularization is an interesting outcome to be considered in future studies and that will most likely contribute with even more interesting diagnostic and/or follow-up information.
In line with points 5 and 6 we have re-written the first paragraph of the discussion section (line 250).
“In the diagnosis of finger pulley injuries, the imaging modality that provides the most information is the high-resolution ultrasound. The following ultrasound signs have been considered for complete pulley ruptures: hypoechoic swelling of the pulley, a dynamic increase in flexor tendon-phalanx gap (equivalent to TBD), flexor sheath effusion and occasionally hyperintense colour doppler [22]. However, to date, the indirect measurement known as TBD has been the most widely considered in scientific literature [3,11–15]. As it is an indirect measure, the specific clinical context is of great value to avoid drawing erroneous diagnostic conclusions. To approach this matter, we here discuss our results in different sub-sections to assess whether the ability to determine finger grip force objectively, the type of grip used when measuring it, the presence or absence of pain, and TBD obtained by ultrasound can together provide relevant information in the diagnosis and/or follow up of isolated A2 or A4 pulley ruptures.”
Results
Point 4. Did the author assess the BMI among four group?
Response: This is a very good question. During the study design process, we also asked ourselves whether it should be important to include it as a variable. We finally decided not to consider the BMI as it has been considered as a bias in a recent study about pulley biomechanics in rock climbers. We have re-written the following paragraph in the discussion (section: Grip Strength Measurement) (line 279):
“As for the formulas used to obtain the finger grip strength deficit value, in the current study the absolute strength value was not divided by the body mass to obtain the relative strength, as this penalizes heavier rock climbers [25]. Furthermore, body mass index was not taken into consideration as in a recent retrospective study it was demonstrated that it is not associated with injuries having occurred within the previous six months as the body can adapt to the load despite high body mass index [26]. On the other hand, in the formulas used by Schöffl et al. [6], the dominant hand was considered, which we did not. In further studies both should be considered when choosing the formulas to measure finger grip strength, avoiding relative force measurements and considering whether the hand measured is the dominant one. “
Discussion
Point 5. This study demonstrated the usefulness of indirect evaluation of A2 or A4 injury (to assess the tendon-bone distance). I think that it seems to be possible to detect A2 or A4 rupture (anatomical rupture) directly with ultrasound. Are there any references related with direct evaluation of A2 or A4 rupture by ultrasonography?
Response: we agree with you, that there are other US signs that could be also relevant. However, as we aimed to compare quantitative strength data to quantitative tendon-phalanx distance (aka TBD) data, we decided not to use qualitative US signs.
Also, by only considering pain upon palpation, TBD and grip strength, the present study has found that several clinical presentation patterns exist. Some are subclinical while others clinical. It would be very interesting that future studies consider other qualitative US injury signs to the different clinical presentations hereby found (considering pain upon palpation, TBD and grip strength) to see if these other US signs can elucidate which patterns are clinically relevant and require a specific management and which are not.
Although it was not an aim of the present study, we found a study mentioning other US signs of complete pulley ruptures, while another describing direct US signs in partial ruptures.
In line with points 6 we have added this to the discussion section (same modification specified in point 3) (line 250):
“In the diagnosis of finger pulley injuries, the imaging modality that provides the most information is the high-resolution ultrasound. The following ultrasound signs have been considered for complete pulley ruptures: hypoechoic swelling of the pulley, an dynamic increase in flexor tendon-phalanx gap (equivalent to TBD), flexor sheath effusion and occasionally hyperintense colour doppler [22]. However, to date, the indirect measurement known as TBD has been the most widely considered in scientific literature [3,11–15]. As it is an indirect measure, the specific clinical context is of great value to avoid drawing erroneous diagnostic conclusions. To approach this matter, we here discuss our results in different sub-sections to assess whether the ability to determine finger grip force objectively, the type of grip used when measuring it, the presence or absence of pain, and TBD obtained by ultrasound can together provide relevant information in the diagnosis and/or follow up of isolated A2 or A4 pulley ruptures.”
Point 6. I think the authors need some work on the discussion. If the authors compare other studies to those that are only directly comparable to their own, I believe this study will have a much more powerful and clear paper.
Response: although we agree with this comment, we also believe that the discussion is already lengthy enough. We intended to discuss studies methodologically directly comparable to ours so we could compare them more accurately. All in all, we have added new references and text to the discussion addressing direct US signs and BMI. We hope that this satisfies the comment in point 6 without making the discussion too lengthy.
Also, with the aim to improve discussion readability we have unified the second and third sections of the discussion, as both answered to the first objective of the study.
Point 7. What is the clinical relevance from the result derived from this study for the treatment of A2 or A4 rupture?
Response: we did summarize so in the conclusion, however we have included the following paragraph in the discussion section, before study limitations (line 351):
“Thus, in A2 or A4 complete pulley ruptures, one finger crimp grip strength measurement is a useful evaluation tool to obtain a more complete diagnosis, along with other well-known diagnosis methods as clinical outcome (mechanism of injury, pain perception, range of movement of the finger by Buck-Gramcko score [32], motion pattern, any other) and ultrasound scanning. Also, grip strength measurement is relevant during the patient follow up, as improvement in strength could be a useful indicator in patient progress [33].”